# Epigallocatechin-3-Gallate Attenuates Myocardial Dysfunction via Inhibition of Endothelial-to-Mesenchymal Transition

**DOI:** 10.3390/antiox12051059

**Published:** 2023-05-07

**Authors:** Sejin Kim, Hyunjae Lee, Hanbyeol Moon, Ran Kim, Minsuk Kim, Seongtae Jeong, Hojin Kim, Sang Hyeon Kim, Soo Seok Hwang, Min Young Lee, Jongmin Kim, Byeong-Wook Song, Woochul Chang

**Affiliations:** 1Department of Biology Education, College of Education, Pusan National University, Busan 46241, Republic of Korea; kimsejin1002@naver.com (S.K.); jjksd@naver.com (H.L.); kimran2448@naver.com (R.K.); huytw01@naver.com (M.K.); 2Institute for Bio-Medical Convergence, Catholic Kwandong University International St. Mary’s Hospital, Incheon 22711, Republic of Korea; moonstar3636@naver.com (H.M.); 91seongtae@gmail.com (S.J.); blue_expanse@naver.com (H.K.); 3Department of Biochemistry and Molecular Biology, Graduate School of Medical Science, Severance Biomedical Science Institute and Brain Korea 21 Project, Yonsei University College of Medicine, Seoul 03722, Republic of Korea; ksang203@yonsei.ac.kr (S.H.K.); hwangss@yuhs.ac (S.S.H.); 4Chronic Intractable Disease Systems Medical Research Center, Institute of Genetic Science, Yonsei University College of Medicine, Seoul 03722, Republic of Korea; 5Department of Molecular Physiology, College of Pharmacy, Kyungpook National University, Daegu 41566, Republic of Korea; vetmedic@knu.ac.kr; 6Department of Life Systems, Sookmyung Women’s University, Seoul 04310, Republic of Korea; jkim@sookmyung.ac.kr

**Keywords:** EGCG, myocardial infarction, EndMT, cardioprotection, oxidative stress, inflammation, fibrosis

## Abstract

Cardiac tissue damage following ischemia leads to cardiomyocyte apoptosis and myocardial fibrosis. Epigallocatechin-3-gallate (EGCG), an active polyphenol flavonoid or catechin, exerts bioactivity in tissues with various diseases and protects ischemic myocardium; however, its association with the endothelial-to-mesenchymal transition (EndMT) is unknown. Human umbilical vein endothelial cells (HUVECs) pretreated with transforming growth factor β2 (TGF-β2) and interleukin 1β (IL-1β) were treated with EGCG to verify cellular function. In addition, EGCG is involved in RhoA GTPase transmission, resulting in reduced cell mobility, oxidative stress, and inflammation-related factors. A mouse myocardial infarction (MI) model was used to confirm the association between EGCG and EndMT in vivo. In the EGCG-treated group, ischemic tissue was regenerated by regulating proteins involved in the EndMT process, and cardioprotection was induced by positively regulating apoptosis and fibrosis of cardiomyocytes. Furthermore, EGCG can reactivate myocardial function due to EndMT inhibition. In summary, our findings confirm that EGCG is an impact activator controlling the cardiac EndMT process derived from ischemic conditions and suggest that supplementation with EGCG may be beneficial in the prevention of cardiovascular disease.

## 1. Introduction

Myocardial infarction (MI) is a multifactorial condition caused by persistent coronary artery or intracardiac ischemia that leads to high mortality owing to complications [1,2]. Cardiac remodeling and heart failure induced by acute MI are associated with myocardial fibrosis (MF), which is characterized by myocardial fibroblast accumulation and extracellular matrix deposition [3,4]. During MI, MF initially prevents infarcted cardiac tissue from rupturing, but the continued induction of MF carries a poor prognosis and concurrently develops into heart failure.

Antioxidants are effective substances for reducing cardiovascular disease and are known to inhibit the oxidation of low-density lipoprotein cholesterol or enzymes that produce reactive oxygen species (ROS); they are often found in plant-based foods and extracts [5,6,7]. Green tea (*Camellia sinensis*) may have antioxidant effects and contains many biologically active polyphenol flavonoids, commonly known as catechins, including epicatechin, epicatechin-3-gallate, epigallocatechin, and epigallocatechin-3-gallate (EGCG) [8]. Among these antioxidants, EGCG enters cells through a receptor called 67LR (67-kDa laminin receptor) and inhibits the generation of ROS within cells [9,10]. There is increasing evidence that EGCG is positively involved in circulatory and metabolic processes, such as cardiovascular disease, diabetes, and obesity [8,11,12,13]. This is possible because EGCG affects multiple intracellular signals and interactions to protect cells from inflammation and inflammation-induced oxidative stress [14]. This is an important reason for the anti-inflammatory, antioxidant, and antifibrotic effects of EGCG, a polyphenol component of green tea.

Tissues damaged under conditions of cardiac ischemia and apoptosis undergo remodeling processes that prevent cell loss and activate regenerative processes [15]. Under harsh conditions, such as hypoxia, endothelial cells undergo transformation into mesenchymal cells, such as myofibroblasts, through endothelial-to-mesenchymal transition (EndMT), leading to cardiac dysfunction and MF [16,17,18,19]. Previous studies provided evidence that EndMT can induce and exacerbate cardiac fibrosis after MI and that transforming growth factor-β (TGF-β) or interleukin-1β (IL-1β) regulate this process. Under hypoxic conditions, microvascular endothelial cells undergo the EndMT process based on a paracrine loop through TGF-β/suppressor of mothers against decapentaplegic (SMAD) signaling, contributing to cardiac remodeling and heart failure that promote MF and cardiac cell death [20]. In addition, we demonstrated that EndMT, which induces the inflammatory cytokines IL-1β, tumor necrosis factor α (TNF-α), and TGF-β in endothelial cells, can lead to pulmonary arterial hypertension [21,22]. Multiple mechanisms for EndMT associated with MI were described to date, including induction of lactate via Snail1 lactylation, promotion of secreted frizzled-related protein 3, which inhibits Wnt signaling, or the addition of pigment epithelium-derived factor, which disrupts β-catenin activation and translocation [23,24,25]. It was also reported that not only cardiomyocytes, but also matrix metalloproteinase (MMP)-14 in macrophages, which are adjacent to the myocardium, regulate EndMT through SMAD2-mediated signaling [26]. However, whether EGCG is involved in EndMT in cardiac fibrosis-induced processes following myocardial ischemic injury remains unknown.

In this study, we investigated the effect of EGCG on endothelial cells during EndMT following MI. We also investigated the effects of EGCG on EndMT during MF in ischemic hearts. To the best of our knowledge, our study is the first to show that EGCG can modulate the EndMT process and has a useful therapeutic role in cardioprotection by inhibiting MF.

## 2. Materials and Methods

### 2.1. Chemicals

EGCG (≥95% by HPLC) from green tea was purchased from Sigma-Aldrich (St. Louis, MO, USA) and dissolved in water. The antibodies used for Western blotting were as follows: fibronectin antibody was purchased from BD Pharmingen; CD31, VE-cadherin, β-actin, total- and phospho-cofilin, total- and phospho-cell division cycle (CDC) 42, RhoA, and total- and phospho-smad2/3 antibodies were purchased from Cell Signaling Technology (Danvers, MA, USA); total- and phospho-NF-κB p65 and total- and phospho-inhibitor of NF (I)κBα antibodies were purchased from Santa Cruz Biotechnology (Dallas, TX, USA); and α-smooth muscle actin (SMA) antibody was purchased from Abcam (Cambridge, UK).

### 2.2. Cell Culture and Treatment

Human umbilical vein ECs (HUVECs) were purchased from ATTC (#PCS-100-010) and cultured with the Endothelial Cell Growth Medium-2 BulletKit^TM^ (EGM-2; Lonza, Walkersville, MC, USA) supplemented 100 U/mL penicillin and streptomycin and were incubated at 37 °C with 5% CO_2_. Cells between passages seven and ten were used for experimentation. The HUVECs were stabilized for 16–24 h after seeding. After stabilization, the cells were co-stimulated with 10 ng/mL transforming growth factor β2 (TGF-β2) and 1 ng/mL IL-1β every 24 h.

### 2.3. Cell Viability Assay

HUVECs were seeded in 96-well plates (2 × 10^4^ cells/well) and incubated in EGM-2 medium for 16 h. The cells were then treated with 0, 1, 5, or 10 μM EGCG for 24 h. After this, the cells were incubated with Cell Counting Kit-8 (CCK-8; Dojindo, Kumamoto, Japan) reagent at 37 °C for 3 h and absorbance was measured at 450 nm using a microplate reader.

### 2.4. Western Blot Assay

Proteins were extracted by centrifugation of the cell lysates at 4 °C for 7 min at 12,000 rpm. Protein concentration was determined using the Pierce bicinchoninic acid (BCA) protein assay (Thermo Fisher Scientific, Waltham, MA, USA). Equal amounts of protein (10–15 μg) were separated using 10% SDS-PAGE, and the separated proteins were transferred onto nitrocellulose membranes. Then, the membranes were blocked with 5% bovine serum albumin (BSA) with 0.1% Tween-20 at room temperature for 1 h, followed by incubation with appropriate primary antibodies (see Section 2.1) at 4 °C overnight. After washing with TBS containing 0.1% Tween-20, the membranes were incubated with anti-rabbit or anti-mouse peroxidase-conjugated secondary antibodies (Thermo Fisher Scientific). The bands were visualized using an enhanced chemiluminescence kit (Advansta, Menlo Park, CA, USA) and ImageJ (ver. 1.52v, NIH) software was used for quantification.

### 2.5. Wound Healing Analysis

HUVECs were seeded in 6-well plates (1 × 10^5^ cells/well) and grown until a monolayer formed. HUVECs were starved for 12 h by changing the medium to serum-free medium, and the monolayer was scraped in a straight line using a 200 μM pipet tip. After the cells were washed gently, the medium was changed to one containing TGF-β2 and IL-1β at 0, 1, 5, and 10 μM EGCG. Images were captured at 0 h and after 24 h, and the percentage of the recovery area compared with 0 h was calculated.

### 2.6. ROS Measurement

Oxidative stress was evaluated by measuring ROS production in HUVECs using H2DCFDA (Ex/Em = 492–495/517–527 nm; Invitrogen, Carlsbad, CA, USA), which is a cell-permeable probe. HUVECs were seeded in a 6-well plate (2 × 10^5^ cells/well). After stabilization for 16–24 h, the cells were treated with 0, 1, 5, 10 μM EGCG, TGF-β2, and IL-1β and incubated at 37 °C for 2 h. Then, HUVECs were washed with phosphate-buffered saline (PBS), incubated with 5 μM H2DCFDA in serum-free medium for 30 min at 37 °C without light.

### 2.7. Mitochondrial Membrane Potential Assay

Mitochondrial dysfunction was evaluated based on mitochondrial membrane potential. This was measured using a tetramethylrhodamine, ethyl ester (TMRE)-Mitochondrial Membrane Potential Assay Kit (Ex/Em = 549/575 nm; Abcam). After HUVECs were seeded in 6-well plates (2 × 10^4^ cells/well), the cells were treated with TGF-β2 and IL-1β, as well as 0, 1, 5, 10 μM EGCG, and incubated for 2 h at 37 °C. Then, the medium was changed to medium containing 100 nM TMRE and incubated for 30 min at 37 °C without light.

### 2.8. LDL Uptake Assay

HUVECs were seeded into 6-well plates (5 × 10^4^ cells/well), co-stimulated with TGF-β2 and IL-1β for 5 days to induce EndMT, and treated simultaneously with 0, 1, 5, or 10 μM EGCG for 24 h. HUVECs were starved for 4 h by changing the medium to serum-free medium. After this, the cells were washed with PBS, and then, serum-free medium containing 10 ng/mL FL-LDL was added. Following incubation at 37 °C for 4 h without light, the cells were observed under a fluorescence microscope.

### 2.9. Permeability Assay

HUVECs were seeded into the insert of a 24-well plate with a 0.4 μm pore size insert (1 × 10^5^ cells/insert) and grown until monolayer formation. Then, 200 μL EGM-2 containing TGF-β2 and IL-1β with 0, 1, 5, or 10 μM EGCG was added to the top chambers, while 500 μL EBM-2 was added to the bottom chambers, and incubated for 24 h at 37 °C. Next, 200 μL EBM-2 containing 1 μg/mL fluorescein-5-isothiocyanate (FITC)-dextran was added to the top chamber and 500 μL EBM-2 in the bottom chamber was replaced with 500 μL EBM-2. Dextran concentrations were measured at an excitation wavelength of 490 nm and an emission wavelength of 525 nm using a microplate reader.

### 2.10. Real-Time PCR

Total RNA from the heart tissue was extracted using RiboEX reagent (Invitrogen). One microgram of total RNA was reverse-transcribed using the Revet Aid First Strand cDNA Synthesis kit (Thermo Fisher Scientific, Rockford, IL, USA), according to the manufacturer’s instructions. Real-time PCR was performed using SYBR Premix Ex Taq (Takara, Shiga, Japan) following the manufacturer’s protocol. All reactions were performed in triplicate. The cDNA was amplified using 60 cycles of 15 s at 95 °C, 15 s at 60 °C, and 30 s at 72 °C for each gene. The expression values are presented relative to β-actin values in the corresponding samples. The primers used in this study are listed in Appendix A.

### 2.11. MI Model and EchA Treatment

Animal studies were approved by the Institutional Animal Care and Use Committee of the Catholic Kwandong University College of Medicine (No. CKU 01-2020-009) and by the Association for Assessment and Accreditation of Laboratory Animal Care. All experimental procedures were performed according to the guidelines and regulations for animal care. Twelve-week-old male C57BL/6 mice were divided into MI + PBS group and MI + EGCG group, each numbering 10 mice. The mice were anesthetized via intraperitoneal injection of tiletamine/zolazepam (Zoletile, 30 mg/kg body weight) and xylazine (10 mg/kg body weight), ventilated with a volume-regulated respirator (VentElite 55-7040, Harvard Apparatus, Holliston, MA, USA), and then, had a median sternotomy. The MI model was established by ligating the left anterior descending artery using a 6-0 Prolene suture (Covidien, Dublin, Ireland) and the muscles and skin were sutured using 4-0 Prolene suture. To perform pathological and functional analysis of the EGCG-treated heart, mice were sacrificed and sampled at 1 week or 4 weeks.

EGCG administration was modified based on previous studies that could inhibit MF after MI [27]. The EGCG solution was prepared as a 10-mmole/L stock in 100% dimethyl sulfoxide as a base. EGCG (50 mg/kg) was administered once daily by oral gavage for seven days after MI induction.

### 2.12. 2,3,5-Triphenyltetrazolium chloride (TTC) Staining

To measure and analyze the ischemic area vs. the normal area, isolated hearts were perfused with 1% TTC (Sigma-Aldrich, St. Louis, MO, USA) for 1 h at 37 °C and left in 4% formaldehyde overnight at 4 °C. Heart sections were taken using a digital camera (DIMIS M model, Anyang, Republic of Korea). The infarcted area in the left ventricle was measured using ImageJ software.

### 2.13. Histology and Immunofluorescence

Five micrometer-thick longitudinal heart tissues were cut from apical to basilar and mounted on a glass slide. To define the fibrotic area of heart, Masson’s trichrome stained area was measured according to standard protocol by using ImageJ software. To determine the number of apoptosis and necrosis in the infarcted myocardium, the terminal deoxynucleotidyl transferase dUTP nick end labeling (TUNEL) Assay Kit—BrdU-Red (Abcam, #ab66110) was used according to the manufacturer’s instructions. TUNEL images were blindly captured and counted at least 4 parts of the left ventricle using a virtual microscope (BX51 Dot Slide; Olympus, Tokyo, Japan). Immunohistochemistry was performed to assess the relationship between EGCG and ischemic myocardium. Sections were deparaffinized and incubated with 1% H_2_O_2_ to lose endogenous peroxidase. Heart tissues were blocked for 1 h with a mixture of 1% (*w*/*v*) BSA and 5% (*v*/*v*) horse serum and incubated with Snail (Invitrogen, Carlsbad, CA, USA, #MA5-14801) or CD31 (Sigma-Aldrich, St. Louis, MO, USA) at a ratio of 1:200. After washing 3 times with PBS, the samples were incubated with secondary antibodies with fluorescent dyes (Alexa Fluor 488, #ab150077 at a ratio of 1:250, Abcam) for 1 h at room temperature, mounted with DAPI, and imaged with a confocal microscope (LSM 700, Carl Zeiss, Oberkochen, Germany).

### 2.14. Cardiac Catheterization

Left ventricular (LV) catheterization was performed 4 weeks after MI to measure invasive hemodynamics. A mouse pressure-volume loop catheter (SPR-839; Millar Instruments, Houston, TX, USA, #SPR-839NR) for simultaneously measuring LV pressure and volume was introduced into the LV via the right carotid artery (closed chest surgery) under anesthesia. Ventricular pressure and real-time volume loops were recorded by a blinded investigator and all data were analyzed using LabChart v8.1.5 software (Millar).

### 2.15. Statistical Analysis

Data were expressed as mean ± standard error of the mean (SEM) of at least three independent experiments. The Student’s *t*-test was used to compare the significance of differences between two groups. Comparisons between three or more groups were performed using one-way analysis of variance (ANOVA) with Bonferroni correction. Statistical significance was set at *p* < 0.05.

## 3. Results

### 3.1. EGCG Inhibits EndMT to Maintain Endothelial Cell Properties

Co-treatment with TGF-β2 and IL-1β produces synergistic effects on EndMT induction [28]. We confirmed the possibility that EGCG treatment contributes to the inhibition of fibrosis by inhibiting EndMT from the co-stimulation of TGF-β2 and IL-1β. First, we observed changes in cell morphology when EndMT was induced by co-stimulation with TGF-β2 and IL-1β in HUVECs. While the control group had a cobblestone-like shape, the TGF-β2 and IL-1β co-stimulated group was spindle-shaped, similar to mesenchymal cells (Figure 1A). We then confirmed whether EGCG treatment affects cell viability; however, no significant effect was observed (Figure 1C). HUVECs were treated with EGCG (0, 1, 5, and 10 μM) for 24 h, and simultaneously co-stimulated with TGF-β2 and IL-1β for 5 days. After this, the expression levels of endothelial and mesenchymal cell markers in the cells were measured using Western blotting to confirm the effect of EGCG on the induction of EndMT [29]. As a result, CD31 and VE-cadherin (endothelial cell markers) were significantly decreased, and fibronectin and α-SMA (mesenchymal cell markers) were significantly increased in the TGF-β2 and IL-1β co-stimulated group compared with the control group; however, their expression levels were significantly reversed by EGCG treatment (Figure 1D,E). Additionally, LDL uptake and FITC-dextran permeability assays were conducted to confirm whether the properties of endothelial cells were maintained in TGF-β2 and IL-1β co-stimulated HUVECs treated with EGCG. Compared to the control group, LDL uptake was decreased in the TGF-β2 and IL-1β co-stimulated group, and this was significantly recovered in the EGCG-treated group (Figure 1F). Moreover, compared with the control group, dextran permeability on the monolayer of HUVECs in the TGF-β2 and IL-1β co-stimulated group was increased and significantly recovered in the EGCG-treated group (Figure 1G). These results indicate that treatment of HUVECs with EGCG can inhibit EndMT induced by TGF-β2 and IL-1β co-stimulation, thereby maintaining endothelial cell properties.

### 3.2. EGCG Inhibits Cell Migration by Affecting the RhoA GTPase Pathway

One characteristic of EndMT is improved cell migration ability [30,31]. Therefore, the effect of EGCG on cell migration was confirmed by a wound healing assay. We compared the recovered areas from the same scratch size in each group. The recovered area was increased in the TGF-β2 and IL-1β co-stimulated group compared with the control group; however, it was significantly decreased in the EGCG-treated group (Figure 2A,B). Moreover, during EndMT, the RhoA GTPase pathway, which is related to the cytoskeletal structure and involves actin, is activated [32]. Thus, RhoA expression and phosphorylation of CDC42 and cofilin were confirmed. In this study, RhoA expression levels as well as CDC42 and cofilin phosphorylation were increased in the TGF-β2 and IL-1β co-stimulated group compared to the control group; however, this was significantly lower in the EGCG-treated group (Figure 2C,D). These results indicate that EGCG can inhibit cell migration ability by affecting the structure of the cytoskeleton, including actin, during EndMT.

### 3.3. EGCG Reduces Oxidative Stress and Regulates NF-κB and SMAD Signaling Pathways

Increased ROS production and inflammatory signals due to TGF-β2 and IL-1β co-stimulation promote EndMT by increasing endogenous TGF-β expression [33]. Thus, we confirmed the effect of EGCG on increased ROS production and inflammatory signaling following TGF-β2 and IL-1β co-stimulation. The intracellular ROS level measured using H2DCFDA was increased in the TGF-β2 and IL-1β co-stimulated group compared with that in the control group; however, EGCG treatment reduced this effect (Figure 3A,B). Furthermore, TGF-β1 increases intracellular ROS and causes the destruction of mitochondrial membrane potential [34]. Therefore, we confirmed whether TGF-β2 and IL-1β also induce the destruction of the mitochondrial membrane potential, and whether EGCG affects this using the TMRE fluorescence assay. TMRE fluorescence levels were significantly decreased in the TGF-β2 and IL-1β co-stimulated group compared to the control group; however, this effect was significantly reversed by EGCG treatment (Figure 3C,D). Additionally, ROS can induce EndMT by activating the NF-κB and SMAD signaling pathways [32]. NF-κB and IκBα phosphorylation levels in the TGF-β2 and IL-1β-co-stimulated group were increased compared with those in the control group, and the SMAD2/3 phosphorylation level was also increased compared to that in the control group; however, these levels were significantly decreased in the EGCG-treated group (Figure 3E,F). The results show that EGCG can inhibit EndMT by inhibiting ROS production, which is increased by TGF-β2 and IL-1β co-stimulation.

### 3.4. EGCG Regulates the EndMT Process in Ischemic Myocardium

EndMT refers to a process in which ECs convert to mesenchymal cells, such as myofibroblasts or smooth muscle cells; this process occurs between 3 and 7 days following ischemia [18]. Based on this process, we established a strategy of daily oral EGCG administration for 7 days to control the induction of EndMT after MI. A dose of 50 mg/kg EGCG was set based on previous studies that delayed MF in vivo after MI [27] (Figure 4A).

To determine the effect of EGCG on EndMT in ischemic hearts, we measured EndMT-related gene expression in mouse heart tissue. *Col1a* and *Tgfb1* exhibit a myofibroblast phenotype under ischemia [35], and *Fsp1* and *Snail* are expressed during EndMT-induced cardiac fibrosis [36]. Furthermore, an increase in *Snail* also increases *vimentin* levels, which promotes degradation of the endothelial basement membrane and rearrangement of the cytoskeleton due to increased MMP expression [37]. We also investigated changes in the EndMT process after MI (Figure 4B). Compared with the normal group, the expression of *Snail*, *Fsp1*, *Col1a*, *Tgfb1*, and *vimentin* in the MI group increased 6.2-, 5.0-, 37.1-, 3.1-, and 8.6-fold, respectively. In the EGCG-treated MI group, however, the expression levels of these genes were reduced by more than half of those in the MI group or similar to those in the normal group.

To evaluate changes in protein levels related to EndMT, immunofluorescence staining and immunoblotting were performed. Snail, which was expressed in the ischemic tissue area, was hardly expressed in the EGCG-treated group (Figure 4C). In addition, collagen I and III, known fibrosis-related factors, as well as MMP-2 and MMP-9 involved in remodeling [16,36], were significantly reduced in the EGCG-treated group (Figure 4D,E). Considering these results, EGCG may inhibit mesenchymal transition and MF by regulating the EndMT process for myocardial ischemia.

### 3.5. EGCG Improves Cardiac Function by Controlling the EndMT Process in Ischemic Heart Tissue

We evaluated whether the degree of MI was improved by controlling the EndMT process after MI. A heart sample was obtained and histologically analyzed one week after ischemia, when the formation of granulation tissue was determined by inducing anti-apoptosis and angiogenesis [38]. Staining of the heart with TTC confirmed that the ischemic area was significantly reduced in the EGCG-treated MI group compared to that in the MI group (Figure 5A). Similarly, TUNEL assays showed that the degree of apoptosis was significantly improved as a result of EGCG treatment (Figure 5B). In addition, the degree of MF (29.2%), which was severe and caused by the development of EndMT after MI, was reduced by approximately one-third in the EGCG-treated group (Figure 5C). In contrast, by checking the ratio of endothelial cells that did not metastasize to the mesenchymal phenotype, we found that the number of CD31 positive cells increased by approximately 2.1 times in the MI group treated with EGCG.

Cardiac function assessment was performed 4 weeks after granulation tissue formation when the scar tissue matured. Millar catheterization analysis was used, in which functional evaluation was performed based on the pressure-volume relationship; load-independent parameters, end-systolic elasticity, end-systolic pressure–volume relationship, and load-dependent parameters, including ejection fraction and end-systolic volume, were improved in the EGCG-treated group compared to the MI group (Figure 5E–G). Collectively, these results suggest that EGCG can be used as a potential therapeutic drug for myocardial protection by inhibiting EndMT and controlling cardiac tissue apoptosis and fibrosis.

## 4. Discussion

The regulation of various cells in the heart exposed to ischemic conditions is one of the most important ways to positively control cardiovascular disease. Properly regulating EndMT, a phenomenon that changes endothelial cells that deliver fresh blood and various factors within the heart, greatly contributes to the positive control and treatment of cardiovascular disease [16]. The induction of inflammation in the myocardium by persistent ischemia is a typical pathological feature of cardiovascular disease. The continuous induction or exposure of cells and tissues to inflammatory cytokines, including IL-1β, TNF-α, TGF-β, and/or interferon-gamma, results in endothelial cell dysfunction and mesenchymal cell acquisition [16,39]. Therefore, protocols or drugs for inactivating EndMT in cardiovascular diseases by controlling the inflammatory response were studied. Losartan, an angiotensin II type 1 receptor antagonist, impaired EndMT by inhibiting TGFβ signaling in TGFβ-treated vascular endothelial cells. Based on this process, TGF-β/SMAD signaling is lowered in hypertensive cardiac fibrosis, resulting in increased angiogenesis and reduced fibrosis [40,41]. Furthermore, simvastatin, a clinical lipid-lowering drug, inhibits EndMT by downregulating TGF-β/SMAD signaling in endothelial cells. A proinflammatory lipid known as 1-Palmitoyl-2-(5-oxovaleroyl)-sn-glycero-3-phosphocholine (POVPC) was found in atherosclerotic lesions; however, POVPC-induced endothelial cells treated with simvastatin have decreased Snail-1 and Twist-1 levels by reducing oxidative stress [42]. Altogether, these findings suggest therapeutic potential following changes in the EndMT process through a complex process in endothelial cells. However, effective drugs to reverse EndMT are still lacking. Therefore, we investigated whether EGCG, which controls inflammation in the ischemic heart and induces antioxidant action, is related to changes in the EndMT process.

Here, by treating HUVECs with TGFβ2 and IL-1β, EndMT conditions were induced to confirm whether endothelial cell characteristics were maintained and whether cell migration was inhibited by influencing RhoA GTPase, NF-κB, and SMAD cell signaling. These results were further supported by a mouse ischemic heart model. At the tissue level, EGCG markedly inhibited EndMT under ischemic conditions, inhibited cardiac fibrosis and cell death following EndMT, and improved myocardial function.

The mechanism by which EGCG induces cardioprotection through the inhibition of myocardial cell death caused by ischemia/reperfusion was first reported in 2004; the study [43] examined the inhibition of signal transducer and found that activator of transcription-1, a transcription factor, was controlled by treatment with EGCG. Lin et al. treated cardiac fibroblasts with EGCG in angiotensin II-treated cardiac fibroblasts to mimic MF after MI. At the cellular and tissue levels, EGCG can attenuate endoglin expression and MF via anti-inflammatory effects through the c-Jun N-terminal kinase/AP-1 pathway [27]. These results indicate that EGCG administration after MI can also exert a cardioprotective effect, reduce the MI area, and control cardiac fibrosis by attenuating inflammation. However, there are no studies on the cardioprotective effects of EGCG. To further understand the cardioprotective mechanism of EGCG, we investigated whether EGCG is involved in the potential mechanism of EndMT in terms of protecting cardiac cells after myocardial ischemic conditions.

Catechins are some of the main components in green tea extract, including EGCG, epigallocatechin, epicatechin gallate, and epicatechin; they are active polyphenol flavonoids that play an important role in various bioactivities, including absorption and excretion. Their structural characteristics play an important role in various bioactivities [8,9]. Polyphenolic flavonoids are produced as secondary metabolites in plants to manage environmental stresses, such as UV light, free radicals, and abnormal temperatures, limiting the effects of oxidative stress. This is closely related to the structure characterized by the presence of one or more phenolic groups capable of reducing cell-attacking ROS and various organic substrates and minerals [44]. These properties were studied for the prevention of chronic diseases caused by oxidative stress. Cardiovascular disease regulates various factors, including blood pressure, endothelial function, and blood lipids [45]. An interesting recent study revealed that quercetin, a flavonoid present in many fruits, vegetables, and seeds, can lower the risk of ischemic heart disease by mitigating EndMT factors that can contribute to endothelial dysfunction [46]. In addition, hydroxytyrosol, a major phenolic compound in extra-virgin olive oil, can positively control atherosclerosis by protecting IL-1β-induced endothelial cells, suggesting that regulation of the EndMT process plays a major role [47]. EndMT activation was induced in vitro by treating endothelial cells with TGF-β2 and IL-1β. The combined treatment of TGF-β2 and IL-1β was reported to induce ‘inflammatory co-stimulation’. This is a strong condition that induces the EndMT process by stimulating the microenvironment at the interface between inflammation (IL-1β) and tissue remodeling (TGF-β2). Based on this condition, the changes of inflammatory EndMT process was confirmed by treating EGCG [28,48]. In vitro studies showed that EGCG was able to alter cell morphology (Figure 1A), mechanistically modulate TGF-β, and, at the same time, reduce inflammatory signals while providing full control (Figure 3). In summary, EGCG enters the cell through the 67LR receptor and reduces ROS generation in the mitochondria, which leads to a decrease in NF-κB production [49]. Additionally, EGCG inhibits the SMAD pathway, indicating its involvement in the overall regulation of the EndMT process induced by TGF-β2 and IL-1β (Figure 5H). In addition, EGCG can significantly inhibit the late EndMT markers MMP-2 and MMP-9 in vivo. Similar to this study, in which EGCG was treated, MMP-2 was reduced to control LPS-induced inflammation in rat H9c2 cells [50], and the effect of reducing MMP-9 was also confirmed in a study targeting myocardial reperfusion [51]. These results suggest that EGCG not only has a cardioprotective effect through antioxidant activity, but also has a direct effect on EndMT, which regulates endothelial cell functions and effectively controls MF, thereby playing a key role in the treatment of cardiovascular disease.

This study revealed that EGCG can have a great effect on improving myocardial dysfunction by controlling the EndMT process. However, dose concentration and side effects (anxiolytic or hypoglycemic activity, hypochromic anemia, liver and kidney failure) should be considered in clinically processing EGCG [52], and research for clinical application should be continued in the future.

## 5. Conclusions

We confirmed the important role of EGCG in protecting against MI-induced cardiac injury and dysfunction through EndMT control. EndMT is an essential mechanism for MF induced after MI, and the results of this study suggest that EGCG does not cause a loss of endothelial function at the cellular level but inhibits cell migration by regulating the RhoA GTPase pathway through the inhibition of RhoA expression and the reduction in cdc42 and cofilin phosphorylation, and also decreases oxidative stress to induce inflammatory signaling pathways, such as NF-κB and SMAD. Our data also highlighted that EGCG induces cardioprotection and enhances function through a dramatic increase in EF and LV pressure by controlling the EndMT process after MI in mice. These results imply that the therapeutic and preventive potential of EGCG in cardiovascular disease can be elucidated based on EndMT.

## Figures and Tables

**Figure 1 antioxidants-12-01059-f001:**
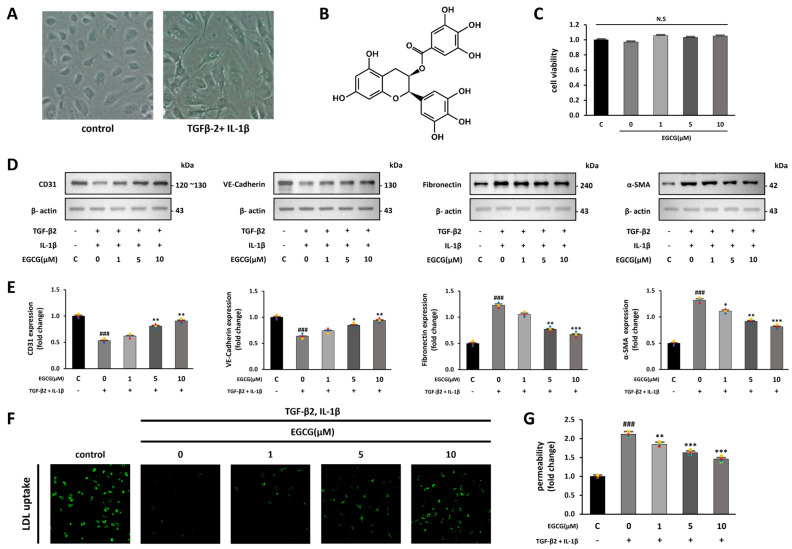
EndMT induction by TGF-β2 and IL-1β and inhibition by EGCG in HUVECs. (**A**) Microscope images of HUVECs in the control and TGF-β2 and IL-1β-treated groups (100× magnification). (**B**) Structural formula of EGCG. (**C**) CCK-8 cell viability assay of HUVECs treated with EGCG. (**D**,**E**) Effect of EGCG on endothelial and mesenchymal cell marker expression levels in HUVECs with TGF-β2 and IL-1β co-stimulation compared with the control group. (**F**) Fluorescence microscopy image of FL-LDL uptake in HUVECs (40× magnification). (**G**) Dextran permeability assay HUVEC monolayer. ^###^ *p* < 0.001 vs. HUVECs without EGCG and TGF-β2 and IL-1β treatment; * *p* < 0.05, ** *p* < 0.01, *** *p* < 0.001 vs. TGF-β2- and IL-1β-induced HUVECs without EGCG treatment. Data were fold-changed for control group (*n* = 3); N.S., not significant.

**Figure 2 antioxidants-12-01059-f002:**
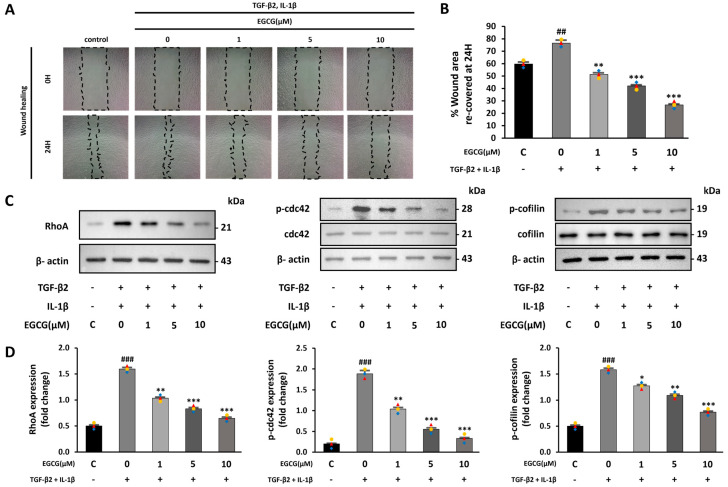
EGCG-induced inhibition of cell migration ability caused by TGF-β2 and IL-1β treatment. (**A**) Microscope images of scratched area observed at 0 h and 24 h after making scratch (10× magnification) and (**B**) percentage of recovered area after 24 h. (**C**,**D**) Effect of EGCG on the expression level of RhoA and phosphorylation level of CDC42 and cofilin in HUVECs co-stimulated with TGF-β2 and IL-1β compared with control group. ^##^ *p* < 0.01, ^###^ *p* < 0.001 vs. HUVECs without EGCG and TGF-β2 and IL-1β treatment; * *p* < 0.05, ** *p* < 0.01, *** *p* < 0.001 vs. TGF-β2- and IL-1β-induced HUVECs without EGCG treatment. Data were fold-changed for control group (*n* = 3).

**Figure 3 antioxidants-12-01059-f003:**
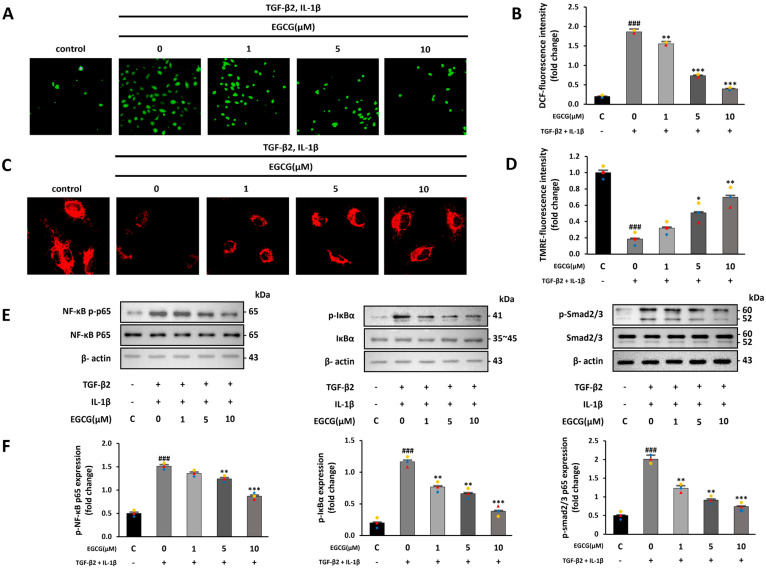
Antioxidant and anti-inflammatory effects of EGCG. (**A**) Fluorescence microscopy image of intracellular ROS produced after TGF-β2 and IL-1β co-stimulation with or without EGCG treatment (40× magnification). (**B**) ROS quantitative analysis. (**C**) Mitochondrial membrane potential measured via TMRE assay (200× magnification). (**D**) TMRE fluorescence intensity. (**E**,**F**) Effect of EGCG on phosphorylation level of NF-κB, IκBα, and SMAD2/3 in TGF-β2 and IL-1β co-stimulated HUVECs compared with control group. ^###^ *p* < 0.001 vs. HUVECs without EGCG and TGF-β2 and IL-1β treatment; * *p* < 0.05, ** *p* < 0.01, *** *p* < 0.001 vs. TGF-β2- and IL-1β-induced HUVECs without EGCG treatment. Data were fold-changed for control group (*n* = 3).

**Figure 4 antioxidants-12-01059-f004:**
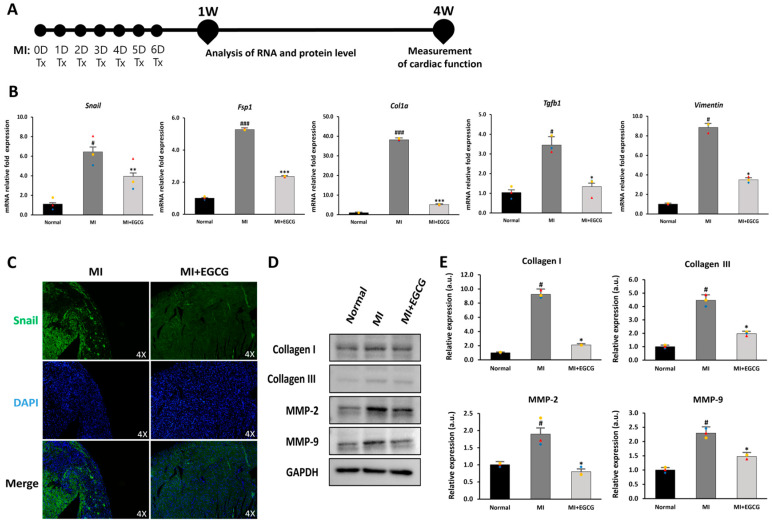
Regulation of EndMT in EGCG-treated MI model. (**A**) Timeline of the animal study. (**B**) Changes in EndMT and EndMT-related gene expression (*Col1a, Fsp1, Snail, Tgfb,* and *vimentin*) according to real-time PCR (*n* = 3). (**C**) Snail expression according to immunofluorescence [green: Snail (EndMT marker), blue: DAPI] (10× magnification). High-quality version: Appendix A. (**D**,**E**) Immunoblot analysis of collagen I, collagen III, MMP-2, and MMP-9 with GAPDH as control (*n* = 3). ^#^ *p* < 0.05 and ^###^ *p* < 0.001 vs. Normal; * *p* < 0.05, ** *p* < 0.01 and *** *p* < 0.001 vs. MI. Data are expressed as X-fold induction compared to normal control. All values are mean ± standard deviation. Statistical significance was assessed by one-way ANOVA.

**Figure 5 antioxidants-12-01059-f005:**
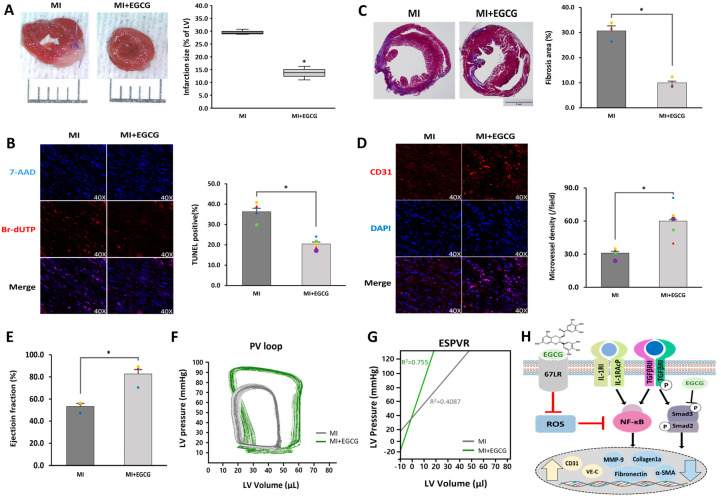
Cardioprotective effects of EGCG on heart tissue after ischemia. (**A**) Infarct size according to TTC staining. Infarct size (%) was calculated as the ratio of the infarcted area (pale) to the risk area (deep-red). Distance between bars = 0.1 cm. *p* = 0.0007 (*n* = 3). (**B**) Apoptotic cell death according to TUNEL assay (TUNEL-positive cells, pink; 7-AAD, blue) performed 1 week after MI (40× magnification). * *p* = 0.0004 (MI: *n* = 3; MI + EGCG: *n* = 4). High-quality version: Appendix A. (**C**) Fibrotic tissue area measured using Masson’s trichrome of cardiac sections 1 week after establishing MI (collagen fibers, light blue; muscle fibers, red). Bar = 2 mm; *p* = 0.0013 (*n* = 3). (**D**) Capillary density according to immunofluorescence staining for CD31 1 week after MI (CD31-positive microvessels, red; DAPI, blue) (40× magnification). * *p* = 0.0035 (*n* = 5). High-quality version: Appendix A. (**E**) Ejection fraction (*n* = 3). (**F**) Representative pressure–volume loop. (**G**) End-systolic pressure–volume relationship (ESPVR). (**H**) A schematic summary of the regulation of EndMT by treating EGCG in ischemic hearts. * *p* < 0.05 vs. MI group. Data are expressed as X-fold induction compared to normal control. All values represent mean ± standard deviation. Statistical significance was assessed by one-way ANOVA.

## Data Availability

The data presented in this study are available in the article and Appendix A.

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
