# Peer review of "Epigallocatechin-3-Gallate Attenuates Myocardial Dysfunction via Inhibition of Endothelial-to-Mesenchymal Transition"

_antioxidants, 2023, doi:10.3390/antiox12051059_

Round 1
Reviewer 1 Report
Research article by Sejin Kim et al. entitled ‘Epigallocatechin-3-gallate Attenuates Myocardial Disfunction via Inhibition of Endothelial-to-Mesenchymal Transition’ deals with an important issue of reducing the burden of cardiovascular diseases. Authors show that EGCG protects endothelial cells (HUVEC) from its transition to mesenchymal. Experiments are performed well however I have some questions for authors before reaching to final decision.
Authors should provide biological replicates as ‘N’ in both invitro and in vivo data.
Authors should make dot bar graph for each represented bar graph data.
Did authors try any or some of the experiments on other endothelial cells? Like HAECs etc. if not please justify that this mechanism of protection of EGCG would be same in other endothelial cells.
Please justify the use of EGCG or green tea in normal wound healing process. Whether EGCG or green tea slows this process in normal conditions or in autoimmune diseases?
If possible, please provide replicates of infarction images along with EGCG group.
Reviewer 2 Report
The manuscript by Kim et al provides evidence in vitro and in vivo that green tea-derived epigallocatechin-3-gallate (EGCG) is protective against myocardial dysfunction, which was caused by endothelial-2-mesenchymal transition.
The study design of combined in vitro and in vivo experiments is well thought through. However, there are some points that should be addressed.
1. It is not clear why the authors used TGF-β2 for the in vitro experiments. Most literature assessed the role of TGF-β1 in MI. This should be better explained in the introduction. In their animal model, the authors determined TGF-β1 levels and not TGF-β2.
2. There is no explanation why the authors combined TGF-β2 with IL-1β for their studies.
3. Either in the cells studies or the animal model, the authors should have determined the concentration- or dose-dependency of EGCG, at least at their end point on EndMT or the cytokine production.
4. The authors determined the expression of MMP2 and MMP9, but they do not provide the rational why these two MMPs were analysed. In the introduction, only MMP14 is mentioned. On Pubmed, over 150 publications that link EGCG to MMP2 and MMP9 can be found. Please correct that.
5. The conclusion has to be rephrased and describes the findings more precisely. E.g. “…but inhibits cells migration by influencing the RhoA GTPase pathway…”, the term “influencing” has to be replaced by a term that describes the role of this pathway more precisely. The same applies to NFκB and SMAD signalling. e.g. “…that EGCG induces cardio protection and enhances function…”, EGCG did not induce cardio protection, it is cardio protective. Furthermore, which function was enhanced by EGCG?
6. The quality of the IF photographs in figures 4C, 5B, and 5D have to be improved. It would be helpful if the authors provide magnified photographs e.g. in the supplementary information.
Reviewer 3 Report
This article reports on interesting studies to cardiac tissue damage caused by ischemia, including cardiomyocyte apoptosis and myocardial fibrosis. The author notes that epigallocatechin-3-gallate (EGCG), can protect the ischemic myocardium. However, its association with the endothelial-to-mesenchymal transition (EndMT) is unclear. The author found that EGCG can regulate proteins involved in the EndMT process of oxidative stress and inflammation-related factors. Further experiments on the myocardial infarction model confirmed that EGCG can regulate cardiomyocyte apoptosis and fibrosis by inhibiting EndMT, and can reactivate myocardial function. Overall, the author's research suggests that supplementation with EGCG can regulate the cardiac EndMT process caused by ischemia, and may be beneficial in the prevention of cardiovascular disease. In spite of its potential interest, there are several weaknesses in the article that must be addressed.
Specific comments
1.The experiment aims to show that myocardial tissue damage after ischemia leads to myocardial cell apoptosis and myocardial fibrosis. However, in the cell experiment, why use transforming growth factor β2 (TGF-β2) and interleukin 1β (IL-1β)? Is it more in line with the clinical condition if the hypoxic environment is used to cultivate? And please explain it is meaningful to choose this cell line.
2. In the presentation of western blot data, normally to explore the message path, the results on the same membrane are used to show different protein expressions, because it can more directly and clearly show the change of the message protein. The original data (Raw data, original Images for Blots) is found to be different pieces, which seems to be relatively inaccurate. If multiple experiments were performed using the same sample, this should be stated.
3. The safe dosage and side effects of EGCG should be mentioned.
Round 2
Reviewer 1 Report
Accept.
Reviewer 2 Report
None
Reviewer 3 Report
This study demonstrates that EGCG supplementation can regulate the ischemia-induced cardiac EndMT process, which may be prevention of cardiovascular diseases.
Thank you for the author's detailed reply, the content has been corrected to perfection.